Understanding park visitors’ soundscape perception using subjective and objective measurement

Ferguson Lauren A. lauren.ferguson@unh.edu 1
Taff B. Derrick 2
Blanford Justine I. 3
Mennitt Daniel J. 4
Mowen Andrew J. 2
Levenhagen Mitchell 5 6
White Crow 7
Monz Christopher A. 8
Francis Clinton D. 7
Barber Jesse R. 5
Newman Peter 2
1 Recreation Management and Policy Department, University of New Hampshire , Durham , NH , United States of America
2 Department of Recreation, Park and Tourism Management, Pennsylvania State University , PA , United States of America
3 Geo-Information Science and Earth Observation, University of Twente , Enschede , Netherlands
4 Mechanical Engineering, Exponent, Inc. , Denver , CO , United States of America
5 Department of Biological Sciences, Boise State University , Boise , ID , United States of America
6 Ramboll Americas Engineering Solutions, Inc. , Milwaukee , WI , United States of America
7 Biological Sciences Department, California Polytechnic State University , San Luis Obispo , CA , United States of America
8 Department of Environment and Society, Utah State University , Logan , UT , United States of America
Aguilar-Perera Alfonso
Electronic publication date: 2024 Jan 31
Publication date: 2024
Volume: 12
Electronic Location ID: e16592
Received 2023 Aug 9; Accepted 2023 Nov 14
Copyright: ©2024 Ferguson et al.
Copyright year: 2024
Copyright holder: Ferguson et al.
License: This is an open access article distributed under the terms of the Creative Commons Attribution License, which permits unrestricted use, distribution, reproduction and adaptation in any medium and for any purpose provided that it is properly attributed. For attribution, the original author(s), title, publication source (PeerJ) and either DOI or URL of the article must be cited.
License URL: https://creativecommons.org/licenses/by/4.0/

Keywords: Soundscapes, Geospatial, Human perception, Parks and protected areas, Noise senstivity, National park

Funding: National Science Foundation CNH 1414171 This work was funded by a National Science Foundation (CNH 1414171) award to Jesse R. Barber, Clinton D. Francis, Peter Newman and Christopher A. Monz. There was no additional external funding received for this study. The funders had no role in study design, data collection and analysis, decision to publish, or preparation of the manuscript.

==============================
Environmental noise knows no boundaries, affecting even protected areas. Noise pollution, originating from both external and internal sources, imposes costs on these areas. It is associated with adverse health effects, while natural sounds contribute to cognitive and emotional improvements as ecosystem services. When it comes to parks, individual visitors hold unique perceptions of soundscapes, which can be shaped by various factors such as their motivations for visiting, personal norms, attitudes towards specific sounds, and expectations. In this study, we utilized linear models and geospatial data to evaluate how visitors’ personal norms and attitudes, the park’s acoustic environment, visitor counts, and the acoustic environment of visitors’ neighborhoods influenced their perception of soundscapes at Muir Woods National Monument. Our findings indicate that visitors’ subjective experiences had a greater impact on their perception of the park’s soundscape compared to purely acoustic factors like sound level of the park itself. Specifically, we found that motivations to hear natural sounds, interference caused by noise, sensitivity to noise, and the sound levels of visitors’ home neighborhoods influenced visitors’ perception of the park’s soundscape. Understanding how personal factors shape visitors’ soundscape perception can assist urban and non-urban park planners in effectively managing visitor experiences and expectations.

Introduction

More than eighty percent of the contiguous United States has elevated sound pressure levels caused by anthropogenic sources (Mennitt et al., 2013). Extensive exposure to noise (defined as unwanted sound) at high levels can negatively affect human health by elevating blood pressure levels, promoting stress, heart disease, hearing loss, and inadequate sleep (Goines & Hagler, 2007; Hammer, Swinburn & Neitzel, 2014). Utilizing U.S. EPA (Environmental Protection Agency) estimates from 1981 and adjusting them to the U.S. population (U.S. Census Bureau, 2010a), 145.5 million people are potentially at risk of developing hypertension because of noise (Hammer, Swinburn & Neitzel, 2014). Urban sound sources such as aircraft, traffic, and people talking have been found to interfere with memory (Benfield et al., 2010), lead to increased stress and lower cognitive ability (Cohen et al., 1980) and cause elevated stress levels in both adults and infants (Cantuaria et al., 2018). Currently more than half (55%) of the world population resides in urban areas and it has been estimated that by 2050, the urban population will grow to 68% (United Nations, 2018). As our society continues to urbanize, the risk for prolonged exposure to loud anthropogenic sounds will rise.

Parks and protected areas serve as places where visitors can find refuge from industrial and community noise (Ferguson, 2018). However, a study found that 63% of protected areas in the United States experience a doubling of background sound levels due to anthropogenic sources and 21% experience a 10-fold increase (Buxton et al., 2017). Thus, protecting natural sounds in parks is important, especially as many visitors to parks and protected areas seek natural sound experiences as a sanctuary from potentially loud and noisy soundscapes they might experience during the course of their daily lives.

Humans have an innate biological association to the natural world (Wilson, 1984) that is also of value for healing the mind and body, as captured by famous nature writers such as Muir (1901) and Thoreau (2016). It has also been documented by many researchers across disciplines (Abbott et al., 2016; Benfield et al., 2014b; Kaplan, 1995; Wilson, 2001). The positive relationship between human health and spending time in nature can promote improved memory retention (Holden & Mercer, 2014) and overall psychological wellbeing (Bratman, Hamilton & Daily, 2012). Experiences in nature can facilitate recovery from mental fatigue (Kaplan, 1995) and a reduction in repetitive negative thoughts (Bratman et al., 2015). Multiple senses are stimulated by natural environments, with natural sound being an important factor (Franco, Shanahan & Fuller, 2017).

In contrast to the negative impacts related to urban noise exposure, there appear to be many positive benefits or psychological ecosystem services related to exposure to natural sounds (Francis et al., 2017; Kogan et al., 2021). Natural soundscapes are important resources to the health and well-being of both humans and wildlife (Francis et al., 2017). For humans, natural sounds improve human cognition (Abbott et al., 2016), enhance positive moods (Benfield et al., 2014a), and increase recovery from stress (Alvarsson, Wiens & Nilsson, 2010). Ferraro et al. (2020) found that park visitors who were exposed to an experimental treatment of increased bird chorus, had improved psychological restoration. A study in Chile’s Coyhaique National Reserve found positive relationships between visitor wellness motivations and soundscape ratings (Ednie et al., 2022). For most visitors to US parks and protected areas and abroad, hearing the sounds of nature and experiencing natural quiet are important motivations for their visit (Ednie et al., 2022; Haas & Wakefield, 1998; Outdoor Industry Foundation, 2011). As visitation to parks increase, so does noise from transportation and other visitors (Levenhagen et al., 2020). As a result, US national parks have plans and polices in place to protect, maintain and restore natural sounds and quiet for both visitor and wildlife health and wellbeing (e.g., National Park Service, 2000).

To understand the complexity of soundscapes, geospatial data is increasingly used to capture the impact of noise and natural sounds on landscapes (e.g., noise impacts throughout the contiguous United States (Mennitt & Fristrup, 2016). Geospatial data depicting sound pressure level have shown how sound from sources such as traffic and construction (Hong & Jeon, 2015; Lee, Chang & Park, 2008; Miller, 2003) are dispersed across landscapes, as well as the pervasiveness of noise pollution in U.S. protected areas (Buxton et al., 2017). These data have also been used to assess racial, ethnic, and social inequalities in relation to noise pollution (Casey et al., 2017). Neighborhood sound levels can be used to answer a variety of questions pertaining to health and the environment. For example, using aircraft noise contours paired with zip code blocks, Correia et al. (2013) identified a correlation between risk of cardiovascular disease in older adults and proximity of their residence to an airport.

Several studies have explored non-acoustic factors that influence perceptions of soundscapes in a park setting (Benfield et al., 2014a; Gale & Ednie, 2020; Kogan et al., 2021; Marin et al., 2011) and in environmental noise assessments (Liu et al., 2013; Miedema & Vos, 1999;1; Schomer et al., 2013). Marin et al. (2011) found that motivations can influence visitors’ perceptions of the park soundscapes. Another study determined that landscape spatial patterns influence soundscape perceptions (e.g., density of vegetation and built environment) (Liu et al., 2013). Noise sensitivity (Benfield et al., 2014a) can also predict visitors’ perceptions of the soundscape they experience in national parks. A recent study conducted in a Chilean national park found that urban visitors who sometimes or often heard anthropogenic sounds perceived those sounds as more acceptable than visitors who never heard those sounds (Ednie & Gale, 2021). These results raise concerns about whether or not visitors are becoming complacent with noise in parks. Another study found that visitors from different urban densities differed in their perception of park soundscapes (Gale, Ednie & Beeftink, 2021). Specifically, visitors from more dense urban areas perceived the soundscape of the Chilean park to be less pleasant. Sounds experienced in daily life influence tolerances to different sources of sound, expectations for the acoustic properties of soundscapes in protected areas, plus motivations for visiting locations where the sonic environment contrasts strongly from that of the routine.

Environmental noise researchers are interested in exploring non-acoustic variables that predict how individuals or communities might respond to noise from airplanes, trains, highway traffic or urban environments (Haac et al., 2019; Miedema & Vos, 1999; Schomer et al., 2013). While sound level meters measure objective physical components of the acoustic environment, human perception of sound varies amongst individuals, communities, and circumstances. The Positive Soundscapes Project, an interdisciplinary study that used qualitative feedback from individuals who participated in urban sound walks, found that soundscape perception is heavily influenced by cognitive and emotional effects (Davies et al., 2013). Researchers have built in this work and identified soundscape descriptors, such as pleasantness, to use in predicting soundscape perceptions (Aletta, Kang, & Axelsoon, 2016). Gale, Ednie & Beeftink (2021), built on methods used by both protected area and urban soundscape researchers to assess soundscape perceptions in a Chilean national park. This research has highlighted the value in using cognitive and affective indicators to measure soundscapes.

Here, we sought to understand what factors influence Muir Woods National Monument (MUWO) visitors’ perceptions of park soundscapes. Based on the outcomes from earlier studies, we predicated that individuals’ motivations (Marin et al., 2011), visitor counts (Stack et al., 2011), and noise sensitivity (Benfield et al., 2014a) would influence visitors’ perceptions of the park’s soundscape. We also hypothesized that the park’s sound level, noise interference, and the sound level of visitor’s neighborhood would predict soundscape perceptions.

Materials & Methods

The analysis presented in this article is part of a larger study in which the primary purpose was to explore the coupling of the natural and human environments through the soundscape, via a paired experiment at Muir Woods National Monument (MUWO). Detailed methods and information about the complete study can be found in Levenhagen et al. (2020). Portions of this text were previously published as part of a doctoral dissertation (https://etda.libraries.psu.edu/files/final_submissions/17621). Field data collection was approved by the US Department of the Interior (Permit #: MUWO-2016-SCI-0001). The survey and social science methodology was approved by the Institutional Review Board of Pennsylvania State University (protocol#: 00004937).

Study area

We conducted this study at MUWO (Fig. 1), the first urban national monument in the United States, located 25 km north of San Francisco, California, and a popular destination for tourists that includes hiking trails throughout 500 acres of coastal redwood trees (Sequoia sempervirens). People are drawn to this park to experience the towering and awe-inspiring old growth coast redwood forest. Visitation to the park has been steadily increasing and in 2017, exceeded one million annual visitors (National Park Service, 2017). Protecting natural soundscapes is a primary management objective at MUWO. Since 2005 the park has supported a variety of soundscape studies (Marin et al., 2011; Pilcher, Newman & Manning, 2009; Stack et al., 2011) that have examined the effectiveness of trail signage to reduce visitor noise (Stack et al., 2011). Due to the findings from Stack et al. (2011) the park now has a sign that states “quiet zone”, in the Cathedral Grove area of the park.

Figure 1 Boundary of Muir Woods National Monument.

This map was created using ArcGIS® software by Esri. Copyright ©Esri.

Experimental design and acoustic measures

We expanded on methods used by Stack et al. (2011) and used educational treatments to designate “quiet days” (treatment) and “control days” during the study period. Stack et al. (2011) tested signage in one small area of MUWO, while our project spanned the entire trail system. We used treatment and control mitigations in weeklong blocks. Additionally, we had rangers enforce the quiet periods. MUWO had one main entrance, which likely made the enforcement more effective than in parks with distributed entrances. During the treatment or “quiet” days, 19 educational A-frame signs (e.g., ‘Enter Quietly’, ‘Maintain Natural Quiet’, ‘What you can do to help natural soundscapes’) were placed along a ∼0.6 km segment of the main trail system. During control days, all educational signs related to maintaining quiet were removed or covered. Additional details related to the experimental design, including a map of the trail, can be found in Levenhagen et al. (2020).

To test the effects of the treatment on background sounds in MUWO, we deployed nine acoustic recording devices (Roland R05) along the same ∼0.6 km segment of the main trail system. These devices were placed 50 m from the main hiking trail. Four other devices were placed more than 100 m from the trail and were not included in our analysis because we were only concerned with the sound levels most likely heard by visitors. The 50th percentile A-weighted sound pressure level (the L50 in dB(A)) was calculated from recordings of each device for each hour (see Levenhagen et al., 2020) for details). To test the influence of the park’s sound level on visitors’ perception of the park soundscape, we paired survey data with the hourly L50 from survey responses based on the hour in which the survey was administered.

Visitor use estimation

For this project, we estimated the number of visitors using the trail during the time that respondents were visiting the park. We used an Automated infrared visitor monitor, TrailMaster (TM1550), located at the entrance to the main trail. Because automatic trail counter estimates can vary with position, angle, etc, a member of the research team observed and manually counted visitors on the trail to calibrate the automated counter. During the study period, the trail counter was calibrated for a total of 12 h (Pettebone, Newman & Lawson, 2010). Manual count calibrations occurred in one-hour blocks, on randomly chosen days throughout the study period. An adjustment factor was calculated by dividing the number of observed visitor pass-by events manually counted during the calibration period by the number of events counted by the automatic monitor. That number was then divided into two, because the monitor location is both an entrance and exit. The mean number of visitors for each one-hour block of time was calculated and multiplied by the final adjustment factor. Visitor estimates were matched with survey data based on the estimated visitor count from the hour of the timestamp on the survey responses.

Survey data

The research team collected a total of 537 surveys between May 9th and 21st, 2016, as visitors were exiting the park. All survey respondents verbally consented to participating in the survey. The survey evaluated the effectiveness of realistic management solutions to improve environmental conditions for wildlife and visitor experiences in MUWO. In addition, we collected data on the tradeoffs visitors would be willing to make in order to achieve a high-quality acoustic experience (Newman et al., 2005). For the purpose of this article, we focused on questions specific to visitors’ perceptions of the soundscape in MUWO that capture pleasantness, noise sensitivity and noise interference (Table 1). In addition, we asked visitors for their home zip code, to identify place of residence.

Table 1 Details on survey questions, response values and how they relate to understanding the visitor and sound perception.

	Question	Value Range	
Motivation	1) To enjoy the natural quiet and sounds of nature
2) To get away from the noise back home
3) Enjoying the peace and quiet
4) Hearing sounds of nature	1 (not at all important)
2 (slightly important)
3 (moderately important)
4 (very important)
5 (extremely important)	
Geographic location	What is your home zip code?	Enter zip code	
Perceptions of the soundscape	
Pleasantness	Visitors hear a lot of sounds, including natural sounds and human-made sounds. Based on your experiences today, how would you rate your pleasantness of the soundscape?	1 (very unpleasant)
2 (moderately unpleasant)
3(slightly unpleasant)
4(slightly pleasant)
5 (moderately pleasant)
6 (very pleasant)	
Noise sensitivity scale	1) I am sensitive to noise
2) I find it hard to relax in a place that’s noisy.
3) I get mad at people who make noise that keeps me from falling asleep or getting work done.
4) I get annoyed when my neighbors are noisy.
5) I get used to most noises without much difficulty (reverse coded).	1 (strongly disagree)
2 (disagree)
3 (slightly disagree)
4(slightly agree)
5(agree)
6 (strongly agree)	
Noise interference	Based on your experience today, how well were you able to hear natural sounds?	1 (almost always clearly without interference from human-made sound)
2 (usually clearly without interference from human-made sound)
3 (sometimes clearly without interference from human-made sound)
4 (usually with interference from human-made sound)
5 (almost always with interference from human-made sounds)	

Pleasantness

For this study, we wanted to test a broad scale that incorporates a positive, well understood attitude towards sound, referred to as pleasantness, which has been found to be an important indicator in measuring urban and rural soundscape perceptions (De Coensel & Botteldooren, 2006).

Noise sensitivity scale

A shortened field version of the noise sensitivity scale (NSS) can be used to measure individuals’ response to noise in their everyday lives and has been empirically validated (Benfield et al., 2014a). We calculated the NSS score after reverse coding one of the items, “I get used to most noises without much difficulty”, to create an overall noise sensitivity score for each respondent. We summated items from the noise sensitivity scale. Lower values indicate a decreased aversion and higher tolerance to noise and higher values indicate increased aversion and lower tolerance to noise.

Noise interference

We developed a measure to investigate respondents’ self-report of how often noise interfered with hearing natural sounds or the degree to which natural sounds were masked by anthropogenic sounds. The higher the value, the more interference from human-made sounds the respondent reported experiencing in MUWO.

Natural sound motivation

Recreation experience preference (REP) scales are used to measure park visitors’ motivations or the desired outcomes they seek in a park or protected area (Manfredo, Driver & Tarrant, 1996). For this study, we were only interested in REP scales related to natural sounds. Prior to calculating the natural sound motivation variable, the four separate items were tested for reliability (Table 2). The natural sound motivation variable was created by summing the four motivation questions related to natural sounds. Internal consistency of the items was assessed using Cronbach’s alpha. The reliability analysis indicated an acceptable level of internal consistency (α = .859) (Vaske, 2008).

Table 2 Reliability analysis and descriptive statistics from independent and dependent variables.

Variables	Mean (sd)	
Single item measures		
Hourly L50	41.36 (1.51)	
Visitor use estimate	214 (68.60)	
Pleasantness	5.24 (1.00)	
Noise Interference	2.37 (1.07)	
Motivation α = .859		
To enjoy the quiet sounds of nature	4.08 (1.01)	
To get away from the noise back home	3.60 (1.30)	
Enjoying the peace and quiet	3.84 (1.08)	
Hearing sounds of nature	3.84 (1.08)	
Noise sensitivity scale α = .808		
I am sensitive to noise	3.76 (1.59)	
I find it hard to relax in a place that’s noisy.	4.48 (1.35)	
I get mad at people who make noise that keeps me from falling asleep or getting work done.	4.41 (1.45)	
I get annoyed when my neighbors are noisy.	4.39 (1.28)	
I get used to most noises without much difficulty (reverse coded).	3.27 (1.27)	

Visitors’ neighborhood sound level

We obtained acoustic data from Mennitt & Fristrup (2016), which approximates the existing L50 sound level at 270 m resolution across the United States during a typical day. We calculated the neighborhood sound level based on the boundary of respondents’ home zip code. For each visitor’s home zip code, the mean sound level was obtained by calculating an average sound level from all grid cells within the zip code. Zip code boundaries were obtained from the United States Census data (U.S. Census Bureau, 2015) and matched to zip codes reported by visitors. A total of 441 unique zip codes were reported by survey respondents. We also eliminated visitors who resided internationally from this analysis. Of these 372 zip codes matched with the boundary shapefile obtained from the Census Bureau and were used for the remainder of the study. We discarded unmatched zip codes as these may have been entered incorrectly. In addition to understanding neighborhood acoustic environments, we used zip codes summarized by state and metropolitan area to better understand where people came from to visit the park. Data on metropolitan areas were obtained from the Census Bureau to identify urban-rural areas where Urbanized Areas (UAs) are defined as areas with 50,000 or more people and Urban Clusters (UCs) are areas with at least 2,500 and less than 50,000 people (U.S. Census Bureau, 2010b). We performed all geospatial tasks in ArcMap 10.4 (Environmental Systems Research Institute (ESRI), 2011).

Data analysis

Given the potential for spatial autocorrelation in the relationship between perceptions of soundscape pleasantness and neighborhood zip code sound levels, in preliminary models we used the fitme function in the spaMM package (Rousset & Ferdy, 2014) in R (version 4.0.4) to incorporate an exponential spatial correlation structure using the Matérn correlation function (e.g., Senzaki et al., 2020; Wilson et al., 2021).

For formal multiple linear regression model selection, we began with a model with neighborhood sound level as the single predictor for soundscape pleasantness and sequentially added additional predictor variables (Table 3). Models with additional variables were retained over the previous hypothesized model if the fixed effects had a p-value < 0.05 and if the Akaike Information Criterion (AIC) was reduced by > 2 from the previously model (Table 3). To test the influence of the experimental quiet signs on pleasantness, we dummy coded this variable (0 = quiet, 1 = control). We also ran an independent samples t-test to examine the difference between the mean L50 heard by visitors who experienced the treatment signs and the control using IBM SPSS Statistics (Version 27). We confirmed the final linear model met model assumptions by visually inspecting diagnostic plots and also found no issues of multicollinearity among predictors using the check_collinearity function in the performance package (Lüdecke et al., 2021). Model selection resulted in a model where pleasantness was explained by neighborhood sound level, noise sensitivity, noise interference, and sound motivation (Table 4 and Fig. 2). We used additional linear models to explore potential predictors of noise interference and noise sensitivity. They were the strongest predictors of pleasantness and we wanted to know more about how they related to the other independent variables in the final model. We created linear models using noise sensitivity as a dependent variable and all of our hypothesized independent variables. We did the same for noise interference.

Table 3 Model selection for pleasantness.

Model	Model equation1	AIC	
M1	Pleasantness ∼Neighborhood sound level	3,405.89	
M2	Pleasantness ∼Neighborhood sound level + noise sensitivity	3,398.69	
M3	Pleasantness ∼Neighborhood sound level + noise sensitivity + noise interference	3,318.29	
M4	Pleasantness ∼Neighborhood sound level + noise sensitivity + noise interference + sound motivation	3,315.47	
M5	Pleasantness ∼Neighborhood sound level + noise sensitivity + noise interference + sound motivation + quiet v. control	3,316.57	
M6	Pleasantness ∼Neighborhood sound level + noise sensitivity + noise interference + sound motivation + hourly L50	3,316.13	
M7	Pleasantness ∼Neighborhood sound level + noise sensitivity + noise interference + visitor count	3,314.03	

Table 4 Final linear model for pleasantness (transformed). Pleasantness∼neighborhood sound level + noise sensitivity + noise interference + sound motivation.

Fixed effects	Estimate	SE	t	P	
Intercept	112.89	12.72	8.88	<0.001	
Neighborhood sound level	−0.50	0.22	−2.30	0.022	
Noise sensitivity	−3.55	1.05	−3.36	<0.001	
Noise interference	−9.54	1.02	−9.37	<0.001	
Sound motivation	2.37	1.08	2.19	0.030	

Figure 2 Marginal effects using the final model (Pleasantness neighborhood sound level + noise sensitivity + noise interference + sound motivation).

(A) Marginal effect of noise sensitivity on pleasantness; (B) Marginal effect of noise interference on pleasantness; (C) Marginal effect of neighborhood sound level on pleasantness; (D) Marginal effect of sound motivation on pleasantness.

Results

Descriptive statistics

The overall mean for hourly L50 was 41.36 dBA (Table 2). Visitors who walked the trail during the quiet treatment (n = 212, M = 41.19 dB(A)) heard a slightly lower and significantly different sound level (t =  − 2.43, p = 0.016) than visitors who walked the trail during the control (n = 159, M = 41.60 dB(A)). We also estimated the number of visitors using the trail during survey respondents’ visit to MUWO. The mean number of visitors on the trail was 214 visitors (SD = 68.60).

For the measure of soundscape pleasantness, the mean score was 5.24 (6-point scale), meaning that the sample on average rated the soundscape as pleasant. Results from the noise sensitivity scale indicated that there was relatively high internal consistency within items (α = .808). To create an overall noise sensitivity score for each visitor, the items were summated. The minimum score was one (low noise sensitivity) and the highest score was six (high noise sensitivity). The mean noise sensitivity score for visitors was 4.10, meaning that the sample of visitors trend towards being sensitive to noise. For noise interference, the mean score was 2.37 (5-point scale). This means that on average, visitors were able to hear natural sounds usually or sometimes clearly without interference from human-made sounds.

The top ranked motivation for visiting MUWO was “seeing the redwoods” and the second was “appreciating the scenic beauty”. The third most important motivation for visiting the park was “to enjoy the natural quiet and sounds of nature”, with a mean rating of 4.08 (on a scale from one to five) (Table 2). Most visitors rated “hearing quiet and sounds of nature” as very important to their visit. To better understand how visitors’ motivations related to soundscape pleasantness, the motivation items related to sound were combined into one motivation score. Overall, these items have a relatively high internal consistency (α = .859). The mean score for the combined sound motivation is 3.83 on a 5-point scale, meaning that on average, visitors rated items related to hearing natural sounds as important to their visit to MUWO.

Sample characteristics and neighborhood sounds level

During May 2016 we found that 82% of the visitors to MWUO were from the United States, 15% were international and 3% did not specify their place of residence. Within the United States, visitors came from 46 different states, with the majority coming from California (30%). Twelve percent of the population were from nearby large urban areas such as San Francisco or Oakland. Moreover, a significant portion of the sample reported being from an urban area (77%), while the other 23% were from rural locations.

The minimum mean L 50of respondents’ zip codes was 31 dBA (the sound level of a soft whisper or light wind) and the maximum mean was 57 dBA (the sound level of traffic). On average, the mean sound level for respondents’ zip codes was 47 dBA, which is comparable to the sound level of a quiet residential or urban neighborhood during the day. Most visitors (63%) came from a neighborhood where sound levels ranged between 40 and 49 dBA.

Linear model explaining soundscape pleasantness in MUWO

Neighborhood sound level, noise sensitivity, noise interference and sound motivation explain 24% of the variance in soundscape pleasantness (multiple R2 = 0.24). Based on the marginal effects from the model (using the untransformed dependent variable), a 1 dB increase in neighborhood sound level results in a 0.02 decrease in the rating of perceived pleasantness (6-point scale) of the soundscape (Fig. 2C). A one-point increase in noise interference resulted in a 0.41 decrease in pleasantness of the soundscape (Fig. 2B). A one-point increase in noise sensitivity resulted in a 0.14 decrease in pleasantness of the soundscape (Fig. 2A). Finally, a one-point increase in motivation to hear natural sounds resulted in a 0.09 increase in pleasantness of the soundscape (Fig. 2D).

Analysis of predictor variables

We found that neighborhood sound level had a small, but significant, negative influence on noise sensitivity (Table 5). Sound motivation had a small, significant and positive effect on noise sensitivity (Table 5). Sound motivation was also a significant predictor of noise interference, along with quiet v. control days (Table 6).

Table 5 Linear model: Noise sensitivity ∼Neighborhood sound level + noise interference + sound motivation + hourly L50 + visitor count + quiet v. control.

Coefficients	Estimate	SE	t	P	
Intercept	6.12	2.22	2.74	<0.001	
Neighborhood sound level	−0.02	0.01	−2.07	0.04	
Noise interference	0.07	0.05	1.38	0.17	
Sound motivation	0.13	0.05	2.52	0.01	
Hourly L50	−0.04	0.06	−0.70	0.48	
Visitor count	−0.00	0.00	−0.95	0.34	
Quiet v. control	−0.08	0.12	−0.76	0.45	
Notes.

R2 = 0.026.

Table 6 Linear model: Noise interference∼Neighborhood sound level + noise sensitivity + sound motivation + hourly L50 + visitor count + quiet v. control.

Coefficients	Estimate	SE	t	P	
Intercept	−1.87	2.21	−0.84	0.40	
Neighborhood sound level	0.01	0.01	0.84	0.40	
Noise sensitivity	0.07	0.05	1.37	0.16	
Sound motivation	−0.10	0.05	−1.89	0.05	
Hourly L50	0.08	0.05	1.42	0.15	
Visitor count	0.00	0.00	3.74	0.65	
Quiet v. control	0.05	0.12	0.46	<0.001	
Notes.

R2 = 0.096.

Discussion

We assessed both subjective and objective measures of visitors’ park experiences. The mean sound level for the park during the time visitors were in the park was 41.36 dB(A), which is what we would expect in a park where visitors could hear sounds like water running, birdsong, and people walking and talking. We also found that the sound level was slightly lower for visitors who experienced the park when quiet signs were posted. Although the difference in decibels is small, it’s important to consider that decibels are a logarithmic measure. For context, an increase of 1.19 dBA equates to about a 24% loss in visitor’s listening area (Levenhagen et al., 2020). So even a small increase in sound level will decrease visitors’ listening areas. When hourly L50 was examined between treatment and control days with the full dataset (this included all visitor hours during the study period), we found that sound levels were significantly lower when signs were present (see Levenhagen et al., 2020 for more detailed results). Moreover, the sound level does not measure sound source. It’s possible that human caused noise was replaced with animal sounds like birdsong (Levenhagen et al., 2020). We found a combination of different factors influenced visitors’ perception of the pleasantness of the soundscape in a park context. Noise interference, noise sensitivity, motivation to hear natural sounds, and sound level of visitor’s neighborhood were significant predictors of soundscape pleasantness. More objective measures, like the sound level of the park and the number of visitors on the trail were not significant variables in our multiple regression model.

Noise interference

A number of studies have focused on the influence of motorized sounds on soundscape experience (e.g., Benfield et al., 2018; Mace, Bell & Loomis, 2004; Mace et al., 2013; Weinzimmer et al., 2014). Our study expands this research to highlight the impact of anthropogenic sound sources such as voices, speakers playing music, and park maintenance machinery, on negative soundscape experiences. Model results show a significant negative relationship between subjects’ rating of noise interference and pleasantness of the soundscape. This factor had the largest effect on pleasantness in the model (Table 4). As the interference with natural sounds increased, the perception of soundscape pleasantness decreased. Based on previous measures of the MUWO soundscape, visitors talking is the most prevalent anthropogenic sound and has the potential to mask or overpower natural sounds (Stack et al., 2011). Benfield et al. (2010) found that hearing recordings of voices have a negative effect on participants’ ratings of national park scenes. Additionally, the increased volume of voice sounds led to increased ratings of annoyance and negatively affected emotional ratings tranquility, freedom, and naturalness (Pilcher, Newman & Manning, 2009).

Our results suggest that the sound level of the park was not a significant predictor of soundscape pleasantness. Noise interference, rather than the acoustic measure of the environment’s sound pressure level, better explained the perception of the soundscape. When a person interprets a sound, it can be the sound source, rather than the sound pressure level that might elicit a positive or negative interpretation or reaction (Alvarez, Angelakis & Rindel, 2006). The sound level of the park includes both natural and anthropogenic sounds. For example, moving water, a sound source that most people find pleasing, was a dominant sound captured by many of the acoustic recording devices during the sampling period. Noise interference was more accurate in predicting how visitors rate the soundscape. These findings differ from Levenhagen et al. (2020) which found hourly sound level to be a significant predictor for proportional odds ratios for pleasantness. Our results are not conflicting, rather in this article we used linear regression modeling to understand variables that predict pleasantness; thus, the assumptions and results here are different from Levenhagen et al. (2020). Additionally, the dataset used in this article differs slightly from Levenhagen et al. (2020) because we only included visitors’ responses whose zip codes matched with the US Census shapefile.

Another notable finding in our study is that the educational signs or the experimental design were not a significant predictor of soundscape pleasantness. However, in our analyses for noise interference (Table 6), the educational signs (quiet v. control) were significant predictors of noise interference. Although significant, the estimate is still small (0.05), suggesting its slight influence on noise interference. The direction of this relationship is positive, which indicates that when quiet signs were covered, ratings of noise interference increased. Levenhagen et al. (2020), using a similar dataset, found the educational signs (quiet v. control) and actual bird diversity were significant in predicting visitors’ perceptions of bird diversity. When the study area was quieted with the treatment of educational signs, visitors were better able to observe bird diversity. Our findings support the effectiveness of the signs ability to improve visitors’ ability to hear natural sounds with less interference from noise.

Noise sensitivity

We found noise sensitivity to be a significant predictor of soundscape pleasantness. Specifically, those who were more sensitive to noise found the soundscape to be less pleasant, though the influence of noise sensitivity on pleasantness was not as strong as the influence of noise interference (Table 4). Nevertheless, this relationship is consistent with data from Rocky Mountain National Park. Benfield et al. (2014a), found that park visitors with higher ratings of noise sensitivity rated aircraft noise as less acceptable and rated other human-made noises as more problematic. We used a linear model to learn more about predictors of noise sensitivity.

Neighborhood sound level had a small negative, but significant effect in predicating noise sensitivity. This small, but negative relationship, suggests that noise sensitivity decreases as the neighborhood sound level increases. It makes sense that these two variables would be related and it would be valuable to understand more about why they are related. Ednie & Gale (2021), found that visitors from urban areas who heard more anthropogenic sounds in a Chilean national park rated anthropogenic sounds as more acceptable than visitors who didn’t hear any anthropogenic sounds. The authors question if urban visitors are complacent with noise in parks.

Sound motivation

Motivation to hear natural sounds was a positive and significant predictor of soundscape pleasantness. The relationship between visitor motivations and perception of the soundscape was consistent with Marin et al. (2011), who determined visitors with higher motivations to experience quiet had lower ratings of human caused noise. This also reflects the findings in our additional predictor variable analysis. We determined a small positive relationship between sound motivation and noise sensitivity. The more sensitive a visitor is to noise, the more likely they are to have a higher motivation score for hearing natural sounds.

Neighborhood sound level

Because perception of the soundscape is influenced by more than just the physical measure of sound (Benfield et al., 2014a), it is important to explore individual characteristics that effect soundscape judgments. Within the environmental noise literature, researchers have concluded that people in different communities perceive identical sounds to be either less annoying or more annoying based on their personal norms and attitudes (Gale, Ednie & Beeftink, 2021; Marin et al., 2011). Differing from previous research, this study is the first to explore the relationship between the sound level of individuals’ neighborhood and their perception of park soundscapes.

Our findings suggest individuals’ home sound environment contributes to visitors’ perception of the pleasantness of the park’s soundscape. Specifically, as neighborhood sound level increased, the rating of soundscape pleasantness decreased. These findings align with Gale, Ednie & Beeftink (2021) who found visitors from more dense urban areas to rate the soundscape of a national park, home, and work differently than visitors from less dense urban areas. Moreover, they found a significant negative correlation between urban density and the park soundscape pleasantness. Indicating that as urban density of the visitors home increased, their rating of the park’s soundscape pleasantness decreased.

Additionally, a large portion of our sampled population was from urban areas (population over 50,000). While the survey did not include questions about these variables, the observed trend could be the result of “learned deafness”, when humans and animals become accustomed to noise (Hatch & Fristrup, 2009; Fristrup, 2015). Individuals could be ignoring the sounds around them to block out unwanted sounds or noise. Whether learned deafness in response to irrelevant sounds transfers to learned deafness to relevant sounds is an important area of future research. For instance, might “learned deafness” influence the magnitude of restorative effects from natural sounds? As mentioned earlier, it’s also possible that visitors are becoming complacent with hearing increased noise in parks (Ednie & Gale, 2021).

This trend could also be a result of people living in urban settings reporting higher rates of noise induced hearing loss (Lewis, Gershon & Neitzel, 2013). Many Americans are exposed to harmful levels of noise (Hammer, Swinburn & Neitzel, 2014) and in 2012 it was estimated that 24% of adults experienced hearing loss because of noise exposure (Carroll et al., 2017). Although it was not measured in this study, it is possible that respondents living in noisy urban areas experience higher rates of hearing loss or other disorders and were less likely to rate the soundscape as pleasant.

Planning and management implications

Management of natural soundscapes in protected areas is important for conserving wildlife, and for providing visitors with holistic benefits. Our findings demonstrate how various factors influence the perception of soundscape pleasantness. MUWO designates certain areas of the park as “quiet zones”, and empirical evidence shows that this method is successful in quieting the park (Stack et al., 2011). It is important for other parks, especially those close to urban centers, to adopt similar management techniques. While parks might be quieter than a busy downtown area, it’s important to keep these protected places quiet, so that visitors have the opportunity benefit from the ecosystem services they provide (Ferraro et al., 2020; Gidlöf-Gunnarsson & Öhrström, 2007).

National park units across the United States are taking steps to implement policies that protect natural soundscapes. Findings from this study suggest that other protected area agencies within the United States and abroad could develop plans to protect natural sounds and quiet, thus leading to a quieter protected area soundscape. In a study of perceived restoration experiences in urban parks, Payne (2008) found that visitors’ perception of the soundscape plays a significant role in their restorative experience. Urban parks that can provide experiences that improve the wellbeing of urbanites should design spaces that reduce human sounds. This can be done by creating messaging and associated zones that influence visitors to keep quiet, avoid cell phone use, and mute music. Finally, this study highlights the importance of quiet natural places, such as urban parks. As the United States continues to urbanize, cities should prioritize the development and maintenance of urban parks for the wellbeing of its residents (Larson, Jennings & Cloutier, 2016)

Limitations and future research

Our study suggests that individual exposure to sound can impact perceptions of a protected area soundscape. Additionally, we found a negative relationship between noise sensitivity and the sound level associated with home zip code. It would be valuable to explicitly examine how noise sensitivity varies with typical noise exposure. We used acoustic data from Mennitt & Fristrup (2016), to estimate visitor’s neighborhood sound level. Future researchers could consider adopting other methods for sound mapping. For example, a study conducted in France used a stochastic modeling approach, which considers temporal sound distribution per sound source, to estimate urban sounds (Aumond, Jacquesson & Can, 2018). Moreover, our results combined with evidence from Ednie & Gale (2021) and Gale, Ednie & Beeftink (2021) should elicit research related to complacency for noise in parks. Visitors from louder, denser urban areas seem to rate park soundscapes as less pleasant. This research could be extended to different national parks and urban parks across the globe to validate this trend. If so, this could be problematic for parks that aim to provide restorative, natural soundscapes.

Conclusion

Parks are important for providing natural soundscapes, especially for people living near urban centers where sound levels are the highest. We show that relationships with soundscapes can be complex and that the sound level experienced on a daily basis can influence one’s perception of a park soundscape. We found that individuals from neighborhoods with higher background sound levels rated the MUWO soundscape as less pleasant. This could be a result of learned deafness and/or a comfort in urban sounds that coincide with living in areas with increased sound levels. Moreover, those who experienced increased interference with natural sounds found the soundscape to be less pleasant. Urban park planners can use evidence from this study to inform future research and management related to natural sounds.

Supplemental Information

Supplemental Information 1 Raw data

Click here for additional data file.

Supplemental Information 2 MUWO Script

Click here for additional data file.

Supplemental Information 3 Survey

Click here for additional data file.

We thank the National Park Service and Muir Woods National Monument for access to study areas, A. Pipkin for acoustic assistance, and A. Petrelli, C. Asher, R. Barber, E. Cinto-Mejía, C. Costigan and C. Levenhagen for project set-up and data collection.

Additional Information and Declarations

Competing Interests

Author Contributions

Human Ethics

Field Study Permissions

Data Availability

Daniel J. Mennitt is employed by Exponent, Inc. and Mitchell Levenhagen is employed by Ramboll Americas Engineering Solutions, Inc.

Lauren A. Ferguson performed the experiments, analyzed the data, prepared figures and/or tables, authored or reviewed drafts of the article, and approved the final draft.

B. Derrick Taff conceived and designed the experiments, authored or reviewed drafts of the article, and approved the final draft.

Justine I. Blanford performed the experiments, analyzed the data, authored or reviewed drafts of the article, and approved the final draft.

Daniel J. Mennitt performed the experiments, analyzed the data, prepared figures and/or tables, authored or reviewed drafts of the article, and approved the final draft.

Andrew J. Mowen analyzed the data, authored or reviewed drafts of the article, and approved the final draft.

Mitchell Levenhagen performed the experiments, analyzed the data, authored or reviewed drafts of the article, and approved the final draft.

Crow White conceived and designed the experiments, authored or reviewed drafts of the article, and approved the final draft.

Christopher A. Monz conceived and designed the experiments, authored or reviewed drafts of the article, and approved the final draft.

Clinton D. Francis conceived and designed the experiments, authored or reviewed drafts of the article, and approved the final draft.

Jesse R. Barber conceived and designed the experiments, authored or reviewed drafts of the article, and approved the final draft.

Peter Newman conceived and designed the experiments, authored or reviewed drafts of the article, and approved the final draft.

The following information was supplied relating to ethical approvals (i.e., approving body and any reference numbers):

Our social science work was approved by the Institutional Review Board of Pennsylvania State University (protocol#: 00004937). We did not use camera traps to count visitors. We used TrailMaster Infrared Trail Counters, they detects the infrared wavelength that people emit as they walk past the device.

The following information was supplied relating to field study approvals (i.e., approving body and any reference numbers):

Field data collection was approved by the United States Department of the Interior National Park Service (Permit #: MUWO-2016-SCI-0001).

The following information was supplied regarding data availability:

The raw data and script are available in the Supplemental Files.

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
