# Peer review of "Understanding park visitors’ soundscape perception using subjective and objective measurement"

_PeerJ, doi:10.7717/peerj.16592_

## Round 0.1 · original submission · Major Revisions

Your manuscript has received very interesting feedback, which I kindly request you to carefully incorporate. I strongly recommend following all the recommendations provided.

·

Basic reporting

I believe your paper is well-written, well-structured, and presents very relevant and interesting research with the potential to advance understanding and practice. Your figures are relevant and well labeled and described. The raw data and supplementary materials are helpful and conform to PeerJ policy. The basic reporting in this article is clear and unambiguous, with professional English used throughout. I have made some specific suggestions about minor issues that can improve specificity and clarity for readers. Although you have incorporated a well-developed review of relevant PA soundscape perceptions literature, I believe the topic of your paper requires a wider consideration of the perceptions literature within urban settings and the newer research that integrates consideration of visitors’ acoustic setting experiences at home, work, and in other natural settings, with their perceptions of pleasantness, and other affective quality measures associated with PA soundscapes. I have made several specific suggestions within the text.

Experimental design

Your research methods are well described and accurately represent the rigorous investigation you have performed. I have two doubts, which are both specified in the PDF annotations. First, I think you need to better set up pleasantness. Second, I think you need to explain what you did with the portion of your sample (18%) that did not provide a US Zip Code. Maybe it would be better to exclude international respondents from the sample and focus on the US portion of the original study sample.

The last three paragraphs of your introduction need further refinement. Incorporating the missing literature and perspectives mentioned earlier will help you more accurately define the research gap your research addresses. Please see my specific comments in notes in the annotated PDF attached to this review. Reworking the last paragraph before the Materials & Methods section will also assist readers. For example, you have not presented formal research questions and hypotheses, and this affects the understanding of your research intent. Your abstract is much clearer than the current wording of this paragraph: “In this study, we utilized linear models and geospatial data to evaluate how visitors' personal norms and attitudes, the park's acoustic environment, visitor counts, and the acoustic environment of visitors' neighborhoods influenced their perception of soundscapes at Muir Woods National Monument.” Try to incorporate this level of clarity within the concluding paragraph of the introduction.

Validity of the findings

I have a few minor questions and suggestions regarding the validity of your findings; all of which are detailed in the PDF annotations. The paragraph written between lines 431 to 441 needs reworking. It is confusing. Also, please consider the implications of your results in the context of the missing literature mentioned earlier. This will further improve the validity of your findings and the quality of your future research recommendations.

Additional comments

Thank you for the opportunity to review your article. I believe your research is very relevant and builds substantially on recent PA soundscape research that has taken a more holistic approach to visitor perceptions by considering the impact of their acoustic experiences and settings outside the park context. Overall, your research is solid, well-organized, well-presented and your writing is clear.

There are three main areas of improvement needed. First, your paper is written through a very U.S.-centric lens. I have noted several places where you can reduce this bias, helping to clarify important points you have drawn out of the literature and the potential transferability and value of your research within a wider geographic context. Second, as your approach to understanding visitor perceptions of PA soundscapes integrates consideration of visitor's home acoustic contexts, I think your literature review needs to incorporate the significant body of urban soundscape research, that has focused on understanding soundscape perceptions in home environments, beyond environmental noise focus. Perhaps the World Soundscape Project (Schafer, Truax, etc.) and Positive Soundscape Project (Davies, would be the place to start, followed by more contemporary work, from Axelsson, Aletta, Kang, Kogan, etc., and most recently, research that has explored how home and work acoustic environments interact with PA soundscape perceptions (Gale, Ednie, etc.). Third, the future research needs you have identified are all valid but, after reading and considering the rest of my feedback, I hope you will expand your perspective about future research. I am hopeful that you will agree that your findings are even more valuable in the context of other recent research findings that have integrated consideration of peoples' home acoustic environments with their perceptions of pleasantness in PA soundscapes. I think building on this research will increase the impact of your study and help you identify additional research needs that move the agenda forward.

Reviewer 2 ·

Basic reporting

The paper “Understanding park visitors’ soundscape perception using subjective and objective measurement” is an interesting research that is included, as explained in the manuscript, in a larger study. The introduction is a bit too long with some information that are not directly considered in the investigation. Figures and tables as well are not all necessary and should be higher quality.

Experimental design

The methods are well described a part from the statistical part. I would move part of the results in the methodology and leave the results section only for what you found. The model selection is fine but the post hoc analyses includes the other parameters not included. I may keep only the first model and remove the post hoc ones. I do not think the table with the model is necessary but if you want to add it you can add the AIC values to show the reason why you decide your model. Finally I would present with more details the results about the L50 of the park during quiet and control period. It is a big part of your method but it is missing in the results and in the discussion part. You could add a graph showing the L50 between the 2 periods and check some differences.

Validity of the findings

Some tables and some figures in the results are not necessary (see the review). I would keep the results part more concise and simpler. Apart form the post hoc analyses that I may remove, the results are well presented and this section is clear.

Additional comments

Introduction
Line 56-62 and Line 81-94: you could merge these two parts and move them at the beginning of the introduction. Then you can speak about parks and then about the natural sounds. I would cut the part about the wildlife as it is not related to your paper.
Line 98-99 What do you mean? I would remove the sentence as it is not related to your paper
Line 110-115: I would remove this part
Line 121-136: I don’t think this part is necessary in your introduction as your paper is not specifically about these topics. I would remove this section.
Line 152: “urban environments” after that I would add a reference
Line 155-158: from “Studies that examine the influenc…themselves (references)” I would remove this sentence
Line 160-172: I would shorten the part of the study aim. You do not need to be so specific now as you are reporting all the methods in the materials and methods part. It is enough just a couple of sentences explaining the main aim of your study.

Materials and Methods
Line 174-179: Is it necessary add this in the methods? Maybe you can put at the end of the paper in the section “Field Study Permissions”
Line 181: I would add a map where you show your study area
Line 183: Sequoia sempervirens in italic
Line 185-189: I would remove this part
Line 206-213: Here I would add a map to show where the devices are (you can merge with the study area one). I think here it is not so clear what you did. You compared the L50 of the 13 devices in the trail with other 9 recorders? Why are you not comparing the L50 from the same devices during quiet and control? I think you should explain deeper this part of your methodology.
Line 206-207: we deployed 13 acoustic recording devices (Roland R05). Here you can also add the specifics of the devices (sampling frequency, data format…)
Table 1: “I get used to most noises without much difficulty (reverse coded)” I would add 5) to consider it as your fifth question
Line 246-263 I would put this section before your table 1. Also I would shorten this section as it is already explained in the Table 1. I don’ t think you need to write “We measured pleasantness of the soundscape on a 6-point categorical scale from very unpleasant to very pleasant” for example
Figure 1: I would delete this figure for a map of your study area. This map is very similar to the one of Mennitt & Fristrup (2016) with the locations of the visitors. I think it is more important the map of the park than this one.
Line 296: I would add the version of R you used for the analysis
Line 309-312: I would rephrase this sentence. You can write something like “no issues of multicollinearity among predictors were found using the check_collinearity function in the performance package (Lüdecke,Ben-Shachar, Patil, Waggoner, & Makowski (2021).

Results
The results should follow the same order of the methods but here you start already with neighborhood sounds level. You can present your results about the L50 in the park. You can add a graph showing the differences/no differences between quiet and control. It is a large part of your methods but in the results and discussion is missing. The “visitor use estimation” is also missing in the result.
Line 315: substitute Muir Woods with MUWO
Line 318: Here you are putting figure 3 before figure 2
Line 320-322 is this result necessary ? I would remove it
Figure 3: I would remove it
Line 323-326: here I would add the fact that most of the visitors came from area with medium sound levels. How did you categorize low, medium and high sound level in figure 2?
Line 332: substitute Muir Woods with MUWO
In the title of Table 2 it is written table 1
I would remove Table 3 as you already explained the model selection in the methods.
Line 352-353: Move “Model selection resulted in a model where pleasantness was explained by neighborhood sound level, noise sensitivity, noise interference and sound motivation” in the methods
Figure 4 is too low definition to be clear. Maybe you can put the 4 figures in one column so they will be bigger and easier to be read
Line 367-372: This part should be move in the methodology. Moreover I would not call that post hoc because the test is the same but with different variables. Why did you decide here to consider all the variables? Including the ones you did not use in the model with the Pleasantness?
Figure 5 seems to be not in high definition.

Discussion
In the discussion you are discussed only the results from the model. You can add at the beginning a section where you can discuss also the other results found (Sample characteristics and neighborhood sounds level and descriptive statistics).
Line 431-441: in the model considering the pleasantness you removed quiet vs control. How do you explain the fact that in a different model is one of the main predictors of one of the main predictors of pleasantness?
How do you calculate quiet vs control?
I would remove Figure 5 as the relationship is not very clear and not so significant.
Line 494-513: but in your results the sound pressure level is not a significant predictor of soundscape pleasantness. Did you measure the differences in L50 between quite and control? Did you find differences?

Conclusion
I would shorten the conclusion, no references and just 4-5 lines resuming your findings.

Reviewer 3 ·

Basic reporting

This article addresses a current topic originally and interestingly.
The work is well structured in blocks and a well-defined and relevant research question is presented in the field of soundscape ecology.
The research is well performed although some sections require clarification before complete acceptance.
- Introduction -
In my opinion, the introduction is the weakest section. The information contained is a bit messy. He returns several times to the same idea but attributing relevance to different sound sources and alternating the use of keywords (soundscape, noise and sound level) without it being clear when these levels refer to anthropogenic or natural sounds. For example, sound levels are related to impacts on human health, wildlife even home values in different paragraphs without a logical sequence. Some sentences seem repetitive and create confusion in the differentiation of three basic concepts (soundscape, acoustic environment and noise) when they are related to their foreseeable effects (positive or negative). Nor is the relevance of noise sources (among industrial, community noise and transportation) clear about protected areas or urban parks. The potential relationship between noise and the economic value of housing is mentioned but, on the other hand, nothing is said about studies that also estimate the monetary value of soundscapes in protected areas or urban parks.
Literature references are quite complete, although some recent studies on the subject published in Europe and Asia are missing, which could enrich both the introduction and the discussion section.

Experimental design

- Materials & Methods -
L193: Although the authors cite Stack et al (2011), they are missing how is defined a “quiet area” and its characteristics in their work. It needs clarification.
Part of the study is based on the previous work of Levenhagen et al. 2020, which is cited several times (L205: details related to the experimental design can be found in Levenhagen et al. (2020)). However, Levengehn et al. (2020) is not included in the list of references (cannot be consulted). It is important to be able to consult this work because several sections of the article are justified based on the methodology of Levenhagen et al. (2020).
An important part of the study consists of contrasting the results of the visitors' surveys with their home zip code and the existing L50 sound level published in Mennitt et al. (2016). Models made by Mennitt et al. (2016) do not directly apply the physics of sound propagation, however, this work shows an application of the potential usefulness of developing different types of maps. In Europe, other mapping alternatives have been developed at different scales, considering different sound sources and with a variety of methodologies that could enrich the discussion section about the usefulness of these tools for the evaluation and management of environmental noise and soundscape perception in parks.
The DOI and URL included in Mennitt et al. (2013) in the list of references are not correct. It leads to a different article, authored by Sueur et al. (2008).

Validity of the findings

- Results -
L323-326 Meaning is attached to some SPL values (in dBA) that may be very different in other regions of the world (for instance those levels assigned to heavy traffic).

- Discussion and Conclusion -
It is somewhat paradoxical that several allusions are made to the relevance of sound levels, although the abstract establishes that “subjective experiences had a greater impact on their perception of the park's soundscape compared to purely acoustic factors like sound level of the park itself ”. For example, it is mentioned that "more than 145 million Americans (~44%) experience sound levels that exceed those recommended to protect public health", although it does not indicate what these recommended levels are (which could be compared with the levels obtained in this study).

Additional comments

- References -
Stack et al 2011a and Stack et al 2011b are the same reference. Furthermore, it is cited in 3 different ways throughout the manuscript (2011, 2011a and 2011b).

- Table 1 -
Only qualifier adjectives corresponding to the extreme values of the scale (1 and 5) are shown. Were there qualifiers for the intermediate values (2, 3 and 4) in the questionnaire?

- Table 2 -
It is named as Table 1.
Consider including the complete motivation ranking.

- Visitors' survey -
Question 1 shows six response options (not relevant, not at all important, …, extremely important). However, in L333-335 of the manuscript, the authors say that they use a scale from one to five). Please clarify this point.

---

## Round 0.2 · Minor Revisions

Please, do the suggested revision and re-submit the manuscript with the corrections.

·

Basic reporting

No comment

Experimental design

No comment

Validity of the findings

No comment

Additional comments

The authors have incorporated all of the feedback provided to them in the first round of review. I believe the article is now ready for acceptance. I saw two very minor spelling errors that require revision and have noted both in the PDF. Chile, the country, should be spelled as such, rather than as Chili. One of the author last names was misspelled for one of the in-text citations and should be corrected. I will recommend acceptance, but would ask that these two minor spelling errors be addressed.

Reviewer 2 ·

Basic reporting

The text is quite clear but some sentences are too long. I would shorten the text and keep the information simple and clear. Tables are fine but the figures (especially figure 2) in my opinion need to be better quality for publication. Some results in my opinion are not so relevant with your research (like estimation number of visitors) and I would consider to remove them.

Experimental design

Some methods in my opinion are describing in not the clearest way. I understand that this research is part of a bigger project but sometimes I cannot understand why you did something very similar to previous studies in the same area (like the sound analysis). Moreover, some methods seem very demanding in time and effort (sound analysis and estimated number of visitors) but the results you showed are very little (mean and sd). Maybe it is better to remove part of the method and keep just the part very releavant for your aim.

Validity of the findings

The findings are very interesting but I would consider to change part of the discussion. In your text you affirmed that the people from urban areas consider not pleasant the sound of the park maybe because they 'prefer' anthropogenic noise but at the same time they visit the park because of the quiet and sounds of nature. I think this part needs to be investigted deeper.

Additional comments

Abstract
Line 38-41: I would shorten the sentence
Introduction
Line 125-126: What is their theory about this result? I would specify it.
Line 147: in your results you are not considering the “density of visitors on the trail”
Materials & Methods
Line 150-151: “The analysis presented in this paper is part of a larger study which primary purpose was to explore…(MUWO).
Line 153: Replace “larger” with “complete”
Figure 1: add the version of the software you used
Line 184: move this part when you mention the model
Line 173-194: As I commented in the previous manuscript for me it is hard to follow here your methodology. In my opinion this part should be shortened describing only the data you are showing in the results. Here I understand that you calculated the L50 for each hour for each of the 13 acoustic devices. Then you compared the L50 for the 13 devices during quiet and control? How? Did you consider only the period while the visitors were in the park? Did you analyze the noise even during the night? Then if I understood well you are using for your model only 9 devices within 50 meters of the trail, is it correct? Did you decide that at 100 meters the noise was not heard by the visitors? Why? Why could not use the 13 or the 9 for both analysis?
Line 189-193: if this part is related to the model, you can move the text when you speak about the model
Line 195-213: In my opinion it is quite complicated to follow here your method to estimate the number of visitors. It is very long and not so clear. My main question is what do you need for the number of visitors? Why do you need a such complicated methodology if in the results you are only mentioning the mean number (per day I guess). You have already the questionnaire and the zip codes that can give you an idea of the people in the park. You probably can get an estimation number from the park on the number of visitors without do all these calculations. In my opinion you could remove this part as in the end it is not important for your results.
Table 1: add the motivation questions in the table
Line 270-277: if in the end you remove the “hour of the survey nested within day” I would remove this part. I think it is a bit complicated to follow here your method. I would keep the text short and simple.
Results
Line 296: In table 2 add the unity system of the L50
Line 298: Did you use a t test to compare quite vs control? You did not mention that in your method. Ok that your results are significant but how do we perceive an increase of 0.41 dB (the difference between the means you found)?
Figure 2 in my opinion is still too low quality for publication. Maybe you can put a and b on top and c and d on the bottom zooming more the images.
Line 338: Figure 2c
Line 339: Figure 2b
Line 340: Figure 2a
Line 342: Figure 2d
Discussion
Line 367-368: How can you tell it is a busy trail? Did you compare it with other similar parks? As I said before all this part of the estimation visitor number for me it is not necessary for your aims.
Line 368-369: “Due to high visitation in MUWO…a reservation.”: I would remove this sentence
Line 385-386: I would remove the sentence: “Hong and Jeon (2014) also found a negative relationship between human sounds and pleasantness, but in an urban context”
Line 386-387I would remove in a lab study
Line 428-430: please rephrase I think something is missing in the sentence.
Line 431-433: What is your theory? I think you can cite some examples based on the literature
Line 437-438: please rephrase I think something is missing in the sentence.
Line 442: How do you explain the fact that “most visitors rated hearing quiet and sounds of nature very important to their visit” but at the same time “respondents living in noisier neighborhoods are accustomed to noise and uncomfortable with, or less appreciative of quiet, natural soundscapes”. If most of the people you interviewed were from urban areas how do you explain this result?

---

## Round 0.3 · accepted · Accept

I consider you followed the reviewers' suggestions well.